# Unique Features of Satellite DNA Transcription in Different Tissues of *Caenorhabditis elegans*

**DOI:** 10.3390/ijms24032970

**Published:** 2023-02-03

**Authors:** Juan A. Subirana, Xavier Messeguer

**Affiliations:** 1Department of Computer Science, Universitat Politècnica de Catalunya, 08034 Barcelona, Spain; 2Reial Acadèmia de Ciències i Arts de Barcelona, La Rambla, 115, 08002 Barcelona, Spain

**Keywords:** tandem repeats, satellites, *Caenorhabditis elegans*, non-coding DNA, small RNA, RNA interference, RNA-Seq, non-coding genome, Helitrons

## Abstract

A large part of the genome is known to be transcribed as non-coding DNA including some tandem repeats (satellites) such as telomeric/centromeric satellites in different species. However, there has been no detailed study on the eventual transcription of the interspersed satellites found in many species. In the present paper, we studied the transcription of the abundant DNA satellites in the nematode *Caenorhabditis elegans* using available RNA-Seq results. We found that many of them have been transcribed, but usually in an irregular manner; different regions of a satellite have been transcribed with variable efficiency. Satellites with a similar repeat sequence also have a different transcription pattern depending on their position in the genome. We also describe the peculiar features of satellites associated with Helitron transposons in *C. elegans*. Our demonstration that some satellite RNAs are transcribed adds a new family of non-coding RNAs, a new element in the world of RNA interference, with new paths for the control of mRNA translation. This is a field that requires further investigation and will provide a deeper understanding of gene expression and control.

## 1. Introduction

A large part of eukaryotic genomes is comprised of repeated sequences including satellites. These sequences may influence chromatin folding, as reviewed by Haws et al. [1]. It is also becoming increasingly clear that most of the genome is transcribed including many repetitive regions [2]. DNA tandem repeats (satellites) are present in most eukaryotic species, but their amount and composition varies significantly, even in closely related species. Centromere and telomere repeats have been studied in great detail [3]. These repeats are frequently expressed as RNA transcripts [4], although the role of such RNAs is poorly understood. A thorough study of repeat transcription in the pericentric heterochromatin of *Drosophila* has recently been published [5]; previous studies in Drosophila have also been reviewed [6]. In the case of human centromeric satellites, it appears that alpha-satellite RNA transcripts are involved in centromere–nucleolus interactions [7]. The transcription of telomeric satellites has also been described [8]. A few other repetitive sequences have been found to be transcribed such as mammalian Alu sequences [9]. However, there has been no detailed study on the transcription of the interspersed satellites found in many species.

The free-living model nematode *Caenorhabditis elegans,* for which RNA-Seq data are available [10], is an adequate species for such study; in a preliminary report, we found that satellites have indeed been transcribed in this species [11]. The genome of *C. elegans* has 102 Mb, distributed over six chromosomes. About 2.5% of the genome is represented by satellites [12], but the exact number will depend on how satellites are defined and on what version of the genome they have been determined. Satellites are distributed all over the chromosomes, but their relative amount is lower in the central third of each chromosome [12].

Here, we report a detailed study on the transcription of the satellites in *C. elegans*. We used the RNA-seq results reported by Kaletsky et al. [10] for our study. They reported the data obtained from 27 samples from four different adult tissues: hypodermis, intestine, neurons, and muscle. We analyzed the influence of size and sequence, particularly in intergenic satellites, which have been considered as junk in the past; their transcription varies in different tissues. We conclude that such transcribed satellites may play a role in RNA metabolism. We also discovered some unique properties in satellites associated with Helitron rolling circle transposons.

## 2. Results

### 2.1. Satellites and Satellite Families

Satellites are tandem repeats of a nucleotide sequence, but they are not perfect. The repeat sequence may show some variability and other sequences may be inserted at some position in the satellite. Furthermore, during evolution, some satellites are transposed to other regions of the genome, and related satellites form a family. In the Methods section, we describe the methodology used to find satellites and their families. We used the WB235 genome version of *C. elegans*, downloaded from the UCSC website [13]. In our previous study, we used WS201, an earlier version of the genome [12]. We also changed the search parameters to find shorter satellites with at least four repeats. Thus, we detected 1989 satellites; a list of them is given in Appendix A. Each satellite is characterized by several parameters, starting with its position in the genome: chromosome number, beginning and final coordinates. The satellite length, number of repeats, and repeat length are also given. Additionally, a similarity parameter measures the variability of the nucleotide sequence of all repeats with the same size in the satellite. In the Appendix A Arabic numerals are used to name the six chromosomes in *C. elegans*; in the text, the standard denomination is used: I–V, X. All of the satellites were positioned in the Ensembl gene data file [13] and 534 satellites present in the intergenic regions were selected for further study; thus we excluded all satellites found in introns and exons.

Satellites with the same repeat length and a similar sequence were aligned to build satellite families (Appendix A), as described in the Methods section. A list of all the families we found is given in Appendix A. Each family is characterized by three values: Fam_*a_b_c*. The order in the list of families is given by *a*, starting with those families with the largest number of members. The second value *b* gives the size of the repeat; *c* gives the number of members in the family. When *c* = 1, the only satellite in the family is unique, and there are no other satellites with the same repeat sequence throughout the genome.

### 2.2. General Features of Satellite Transcripts

The results we obtained from the RNA-Seq experiments [10] are reported in Appendix A. Several experiments were reported for each tissue, with a different number of data (Appendix A). In these experiments, no filtering was carried out to select RNAs with a polyA tail, which would limit the results to protein coding genes.

We found 1081 satellites that showed some transcription. We classified them in three groups: 534 satellites were found in the intergenic regions and 40 in the exons, which corresponded to amino acid repeats in proteins; the rest belonged to introns. We selected the satellites in the intergenic regions for further study. The number of hits of each of these satellites in every experiment was determined (Appendix A) and they were added to obtain a total number of hits in each tissue (Appendix A). 

According to the RPKM values (Appendix A), transcription is usually higher in neurons, since the average RPKM value for neurons is 0.37, whereas for the other tissues it is 0.07–0.15. Some satellites show very low transcription. However, it is difficult to determine which data should be eliminated on this basis. In Appendix A, there were 56 satellites with less than 20 RNA-Seq hits, some of which were clearly noise. However, it is not feasible to determine a threshold, since in some cases, all of these hits fall in a single tissue. Alternatively, we can conclude that all satellites with a RPKM value above the average value in a tissue (Appendix A) should be considered as positive proof of satellite transcription. In Table 1, we present the satellites that have the highest transcription in each tissue; these belong to different satellite families. Some fragments of Helitrons also show a high transcription.

We obtained a graphical representation of transcription for all satellites in the four tissues; high resolution figures are given in Appendix A. Three examples are given in Figure 1. In the next section, we discuss the features of the main satellite families. The satellites presented in Figure 1 are an example of the general features exhibited by the transcribed satellites:**Satellites of the same family may be found in both introns and in intergenic regions.** The relative amount of both types of satellites varies in different families. A few examples are given below. Figure 2 presents the case of a family that shows a predominant intergenic localization.**Transcription is usually higher in neurons** (blue), with a few exceptions, which are apparent in Table 1. A clear case is satellite X: 17441768, also shown in Figure 1; this is mainly transcribed in the hypodermis. This satellite is unique: there are no other satellites in the whole genome with the same repeat sequence.**Satellite transcription is not uniform**, as indicated by the peaks apparent in all cases. This is somewhat surprising, since different repeat units in each satellite have a similar sequence. This fact may indicate that small mutations of individual repeats have a strong influence on the transcription of different regions of the satellite.**The pattern of transcription of each satellite is similar in all tissues.** This is clearly apparent in Figure 1; peaks of transcription can be found in approximately the same position in all tissues, although the peaks showed a different magnitude.**Satellites of the same family have a different pattern of transcription,** as shown by the two satellites in the upper frames of Figure 1, which belong to Fam_84_79_3. The third member of this family (IV: 11033628) has a much lower transcription. These differences may be due to the position of the satellite in the genome and to the detailed repeat sequence. A more detailed example is given in Figure 2, where we present the results obtained in a family of seven satellites.

### 2.3. Satellite Transcription in Different Satellite Families

We first discuss a few families with a large number of members. Some of them have a short repeat (less than 20 nt) and do not show any transcription such as Fam_6_11_75, Fam_7_15_70 and Fam_8_16_50; these may be considered as microsatellites.

Satellites in Fam_1_35_172 are mainly found in introns, only 32% of them are in intergenic regions. The satellites from this family are absent in chromosome X and are distributed on all other chromosomes with a similar frequency. This peculiar distribution may indicate some relationship with the biological function of chromosome X. The average transcription of intergenic satellites in this family is rather low compared with other families.

Fam_4_35_115 has the same 35 nt repeat length but quite different properties: it is found in all chromosomes including X; it has a normal average transcription and 50.4% of its satellites are found in intergenic regions. Some of these satellites are found in Helitrons, which will be described below. A peculiar feature of this family is the presence of a large proportion of satellites (49 of the total 115) that have a shorter repeat length (31–34 nt).

We also studied in more detail a few families that presented some peculiarities. Three of these families have very long satellites, predominate in neurons, and are found in several chromosomes: Fam_65_184_4 has very long satellites in intergenic regions, also found in muscle. Two of them may be long terminal introns of small genes. Found in chromosomes I, II, III, and V. In V, it appears as part of gene wrb-1. Fam_84_79_3 has very long satellites in intergenic regions; known as CeRep53 [13]. It is found in chromosomes II and IV. The transcription pattern of two satellites in this family is presented in Figure 1. The pattern of transcription of Fam_42_94_7 is presented in Figure 2. This family has two very long satellites (13 and 38 Kb); they are intergenic and near the non-coding RNAs linc-85 and linc-165. It has also four shorter satellites: three intergenic, one in an intron. Found in chromosomes I, III, IV, and X.

The X:17441768 satellite is the only member of Fam_234_45_1. It is only present once in the X chromosome. It has 134 repeats of 45 nt, with a total length of 6016 nt. The 45 nt sequence is absent in all other chromosomes and elsewhere in X, as judged by a Blast search. It shows 4700 transcription hits in the hypodermis and 123–907 in other tissues. As shown in Figure 1, this satellite is fully transcribed in the hypodermis and partially transcribed in the other tissues. Transcription in the other tissues is limited and selective; only certain regions of the satellite are transcribed: it is obvious that the satellite is not transcribed as a whole. The RPKM values for this satellite are also included in Table 1.

Fam_46_32_6 is part of the Helitron sequences and is described in detail in Section 2.4. It is only present in chromosome II and is mainly transcribed in intestine cells.

### 2.4. Satellites Associated with Helitron Transposons

Helitrons are a unique type of rolling circle transposons that are very abundant in many eukaryotes including *C. elegans* [14,15]. In most species, they are associated with satellites. In *C. elegans*, they show a unique association, which we describe in this section; only six Helitrons are complete elements, with an associated nuclease gene. Other Helitrons only contain part of the sequence and no associated gene. A complete list of a total 4643 Helitron related sequences is available on the UCSC website [13]. A scheme of a whole Helitron with its associated satellites is presented in Figure 3. 

In the genome of *C. elegans*, there were only six complete Helitron sequences, with an internal transposon gene. The central transposon gene is surrounded by two conserved regions, which we named 5′UTR and 3′UTR, together with three satellite regions. Satellites with a repeat length of 15 nt belong to Fam_3_15_136; those with repeat 32 nt belong to Fam_4_35_115. The satellites at the other end, with a repeat of 31 nt, belong to Fam_46_32_6; this satellite family is only found in chromosome II, all of its six members are associated with the transposon genes. No related satellites are found anywhere in the genome. These Fam_46_32_6 satellites have a variable length and are in direct contact with the end of the transposon gene. Some of these satellites associated with Helitrons show a very high transcription (Table 1).

A striking feature of the satellites in Fam_46_32_6 is that they are mainly transcribed in the intestine. Transcription is higher in the region proximal to the 3′UTR. It is also higher than transcription throughout the gene, as shown in Figure 4. On the other hand, the 15 and 32 nt satellites show very little transcription.

In summary, the six complete Helitrons and their associated satellites show two unique properties that cannot be easily interpreted: first, all are found in a short region of chromosome II (Table 2), and second, they are mainly transcribed in the intestine. The large number of incomplete elements scattered throughout the genome contain fragments of the UTR regions and satellites from Fam_4_35_115, but no satellite fragments of Fam_42_32_6. An additional puzzling feature is their localized transcription, as demonstrated in Figure 4.

All of these complete Helitrons are located in chromosome II. Different regions are presented in the orientation of transcription of the central gene. ZK250.10 is considered to be a pseudogene. Only the starting coordinate of each satellite is given. The terminal conserved region may be longer in some cases. All satellites in Fam_46_32_6 have a repeat length of 31 nt. All 32 nt satellites belong to Fam_4_35_115. The 15 nt satellites are very short and irregular. Some of them belong to Fam_3_15_136. The 32 nt satellite at position 856800 is irregular and does not appear in our list of satellites.

## 3. Discussion

Our study clearly shows that many satellites are transcribed, but with our software programs, it is not possible to determine their precise function. We analyze some possibilities below, but first we summarize some of the intriguing features of satellite transcription. The figures that we have presented demonstrate that satellites are not transcribed as a whole; different regions of each satellite are transcribed more efficiently than others. Additionally, the average repeat sequence does not determine the rate of transcription: satellites of the same family have very different transcription features. A clear example is shown in Figure 2. It appears that both the position in the genome and the detailed repeat sequence determine transcription. We conclude that each individual satellite has unique properties. Point mutations in individual repeats may determine the local RNA conformation.

In our previous study [11], we showed some figures of the simulated conformation of transcribed satellite RNAs. We found that different types of satellites give rise to similar structures, with many double stranded RNA branches. An example is presented in Figure 5. Once transcribed, satellite RNA may remain as such in the cell or be degraded into small RNA duplexes that are found widespread in eukaryotes [16]. Many types of small RNAs have been described [17]. Thus, satellite RNAs may have a role as such small RNAs. Different classes of small non-coding RNAs exert their regulatory functions by directly base pairing with mRNA targets to alter their stability and/or affect their translation [18]. Small RNAs have been thoroughly studied in *C. elegans* [19,20]; they play an important role in germline development [21].

Alternatively, a transcribed portion or whole satellite RNA may interact with other RNA molecules [23,24] and either alter or inhibit their function. Interaction with basic proteins is also possible; RNA binding proteins are essential in RNA metabolism [25]. In humans, they are involved in some genetic diseases [26]. These interactions may be either precise, as molecule–molecule interactions, or satellite RNA may trap several molecules, acting as a sponge and eventually leading to phase separation [27]. In that case, an RNA condensate may form. RNA condensates are an object of great current interest: the journal RNA has recently dedicated a Special Issue to this subject [28].

It has also been suggested that RNA, along with RNA binding proteins, might mediate chromatin organization. The distribution of RNA in the nucleus is not homogeneous [29]. Long satellite RNAs may form complex secondary structures that provide unique domains for interaction with specific proteins and other RNA molecules. A single satellite RNA may act as an RNA scaffold either by interacting with multiple copies of the same protein or several different proteins at once. Satellite RNA associated with chromatin modifier proteins may contribute to stabilize RNA compartments in the nucleus [30,31]. This formation of RNA compartments in the nucleus is currently being intensively studied in *C. elegans* [32].

In conclusion, our results demonstrate that many interspersed DNA satellites are transcribed in different tissues. DNA satellites can no longer be considered as a useless feature of the genome. To find out the exact mode of action of these non-coding RNAs, further experimental studies are required. As noted many years ago by Mattick and collaborators [2], the genomes of all studied eukaryotes are almost entirely transcribed, generating an enormous number of non-coding RNAs. Our demonstration that many satellite DNAs are transcribed adds a new family of non-coding RNAs. The eukaryotic genome may indeed be considered as an RNA machine.

## 4. Materials and Methods

### 4.1. Satellites and Satellite Families

We first determined the distribution of satellites and their families in the WB235 genome sequence of *C. elegans* [13]. Satellites were identified with the program SATFIND, described in detail in a previous publication [12]. SATFIND is available online for general use on our website [33]. The program determines the localization of clusters of any short sequence of a prefixed size without internal repetitions and repeated a minimum number of times in regions with a fixed size. In this paper, we used the SATFIND program to identify satellites formed by at least four repeats of any decamer sequence in 800 nt long regions. In this way, repeats of 10–200 nucleotides repeated at least four times can be positioned in the genome, with no upper limit for the number of repeats in the satellite. Most satellites have a regular structure, but there is a significant number that presents variations in repeat length and composition. In order to eliminate the most irregular satellites, we only accepted those that have at least 60% of their repeats with an identical length. A complete list of the 1989 satellites found with these parameters is given in Appendix A.

To compare the satellites, we used Malig, a progressive multiple sequence alignment algorithm that we developed to align satellite repeats and identify families with a related sequence. Its source code has been deposited in Dryad [34]. The program also considers reverse sequences, normalizes the alignment score to the maximum possible value, and selects the cycle permutation with the highest score. Then, the progressive multi-alignment is applied to the matrix of pairwise alignment scores. The process finishes when the score is smaller than a similarity threshold (input parameter), which we set to 0.6. The consensus sequence of the repeat can be calculated taking into account the circularly permuted sequence of all repeats.

Each family is characterized by three values (e.g., Fam_*a_b_c*). The order in the list of families is given by “*a*”, starting with those families with the largest number of members. The second value “*b*” gives the size of the repeat; “*c*” gives the number of members in the family. Unique satellites appear at the end of the list as families with a single member.

### 4.2. Analysis of RNA-Seq Data

We next searched the RNA-Seq hits in this group of satellites using the data published by Kaletsky et al. [10]. These RNA-Seq data contain SRR files with results from 27 different samples of four tissues of *C. elegans.* The number of hits found for each satellite in all these SRR files is given in Appendix A. The transcription for each satellite was calculated by adding the data for all SRR files from the same tissue.

In order to normalize our results, we determined with Blast the RNA-Seq hits that matched a part or an entire satellite with at least 97% accuracy. To avoid lateral artifacts, we added 250 nt to each side of the satellite in the Blast search. As satellites belonging to the same family are very similar, some RNA-Seqs match many satellites. We selected the hit (or hits) with maximum accuracy since it is not possible to know whether one or more satellites are fully transcribed; the number of hits of each satellite per database is given in Appendix A. 

We also computed the RPKM index for normalization: RPKM = number of hits * 10^9^/(length of satellite)/(number of RNA-seq hits)(1)

This index normalizes the number of hits with respect to the length of the satellite and the number of RNA-Seqs of the database. In Appendix A, we present the RPKM values for all satellites in different RNA-Seq data files. The total RPKM values for each satellite and tissue was computed by adding the values of all the data for each tissue (Appendix A).

In order to visualize the transcription along each satellite, we prepared figures in which the ordinate in every nucleotide position represents the number of RNA-Seq hits that cover this nucleotide. As such, the peaks in the figures represent the regions with more RNA-Seq hits, and therefore a higher transcription. In each figure, we represent the data for each tissue with different colors. In the figures, we also added the neighboring 250 nt of the genome sequence at each end of the satellite in order to avoid terminal artifacts. When the ordinate is zero, there is no transcription. All figures are available in Appendix A.

## Figures and Tables

**Figure 1 ijms-24-02970-f001:**
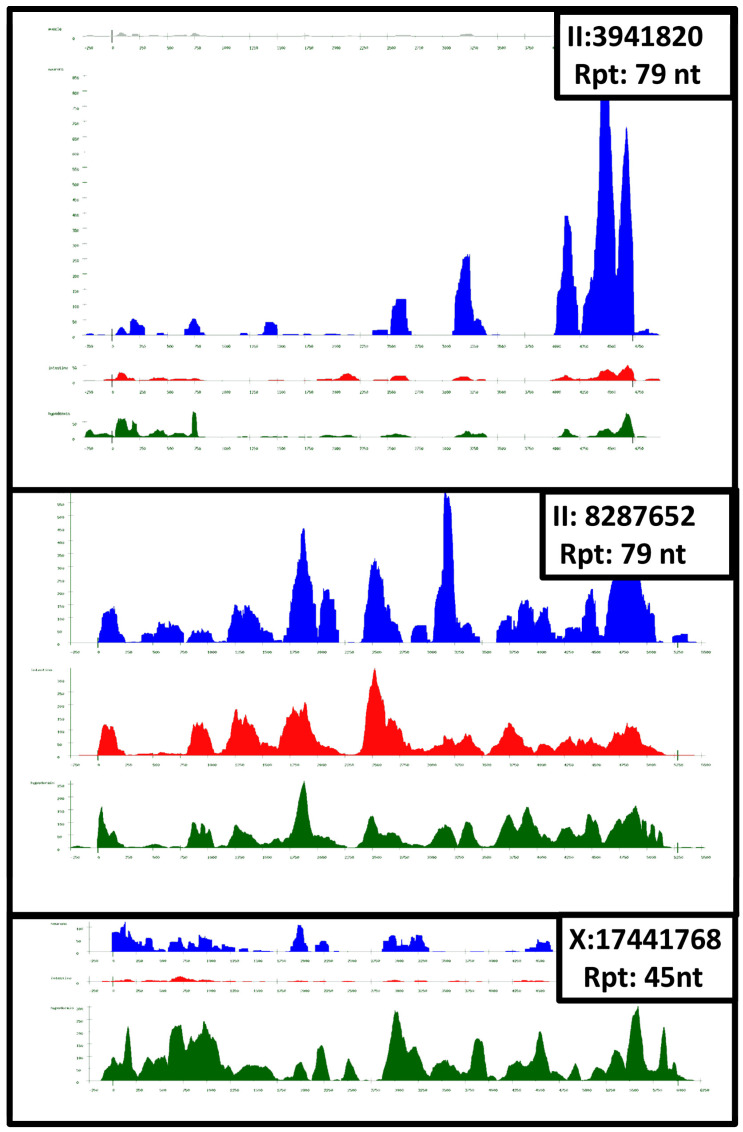
Comparison of transcription in three long satellites (4–6 Kb). Colors in different tissues: grey, muscle; blue, neurons; red, intestine; green, hypodermis. Transcription in muscle is very low in these satellites, it is only shown for the II: 3841820 satellite (upper frame). Further details are given in the main text.

**Figure 2 ijms-24-02970-f002:**
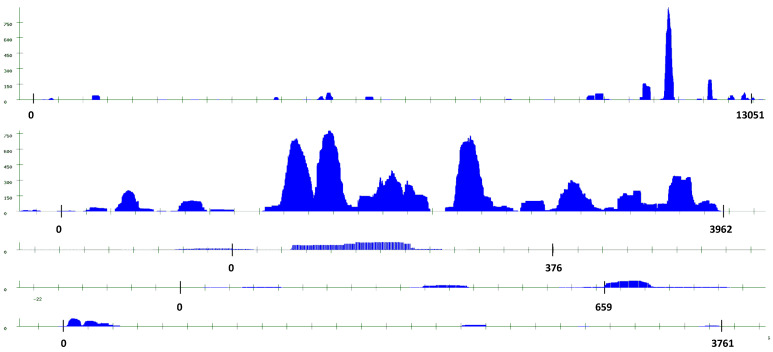
Fam_42_94_7. Examples of the different pattern of transcription in neurons; found in satellites with the same repeat sequence of 94 nt. The origin and end points are indicated for each satellite. Satellites in this family have very different lengths and very different expression patterns. One of the longest satellites we detected (37,806 nt) also belongs to this family. Due to its length, it is not presented here; a high resolution image is available in Appendix A. An additional satellite of this family (I: 10913365) is found in an intron of gene F49D11.4. This family corresponds to CeRep59 in the UCSC Genome Browser [13].

**Figure 3 ijms-24-02970-f003:**
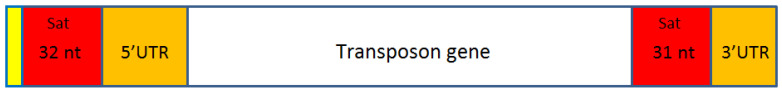
Scheme of a complete *C. elegans* Helitron. All satellites in Fam_46_32_6, with a 31 nt repeat, are part of Helitron1_CE sequences. The whole length of complete Helitrons varies in each case, between 7115 and 8956 nt. It is represented in a simplified way in this drawing. At the left, there is a short satellite with a repeat length of 15 nt (in yellow). Details of each region are given in Table 2.

**Figure 4 ijms-24-02970-f004:**
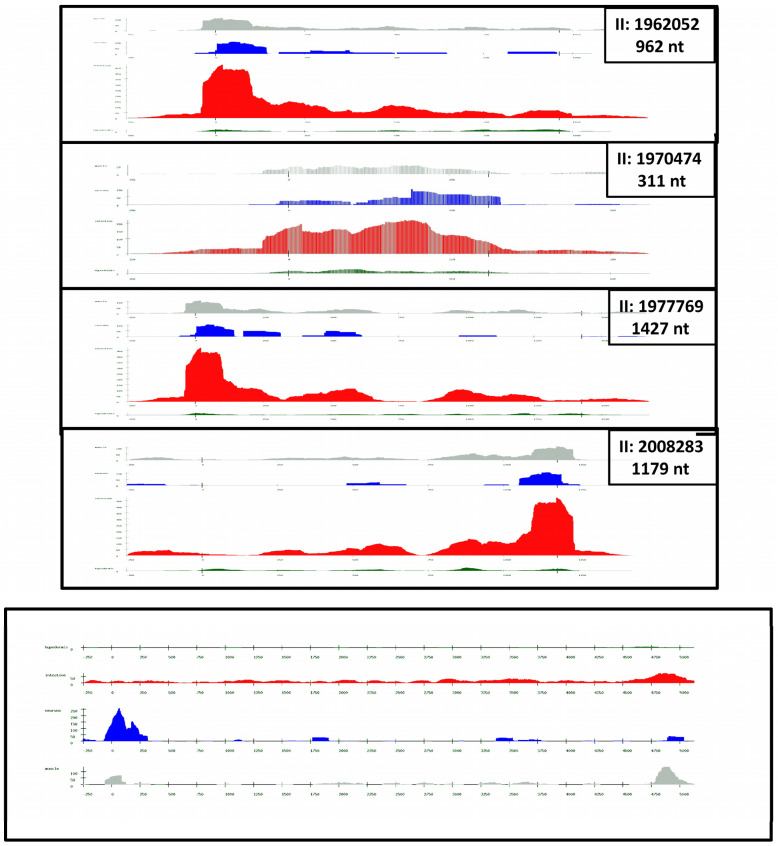
Transcription of 31 nt satellites and one Helitron gene. The transcription of four 31 nt satellites is shown in the upper four frames; the length of each satellite is also given. Note that the orientation of satellite II:2008283 is different from the other cases (Table 2). The images have been vertically compressed for a better comparison, although resolution is diminished. High resolution figures are available in Appendix A. The transcription of the F59H6.5 gene is shown in the lower figure; in the genome, it is found in close contact and before the II: 2008283 satellite; peaks of RNA-seq hits appear at the ends in both cases.

**Figure 5 ijms-24-02970-f005:**
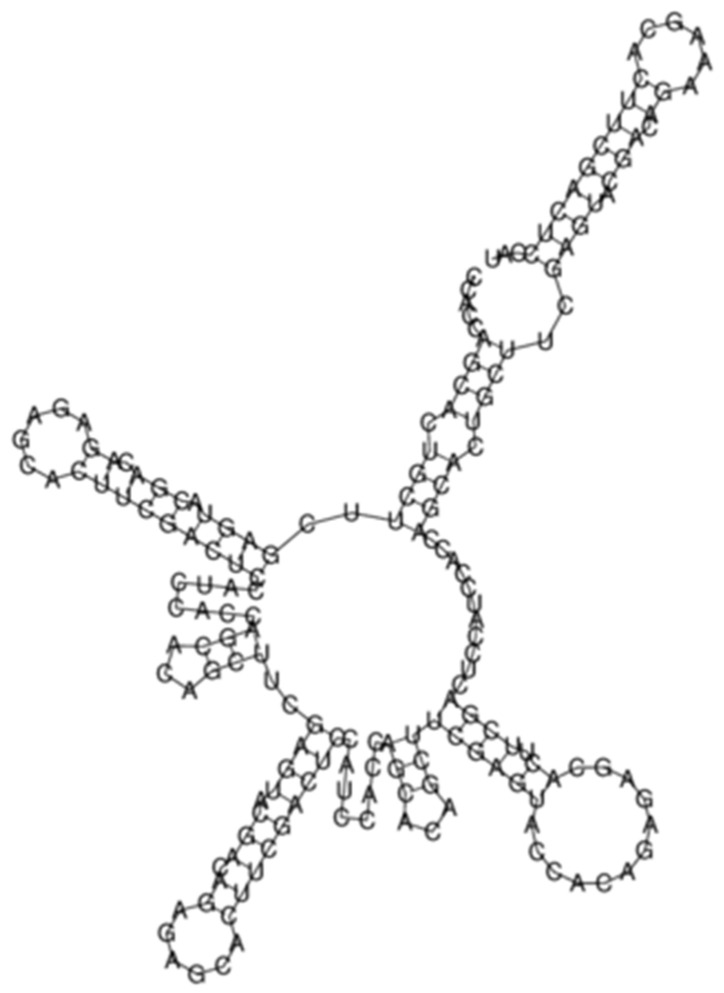
Simulated structure of five repeats of the satellite at position X:17441768, calculated with the RNAfold program [22].

**Table 1 ijms-24-02970-t001:** Satellites with the highest transcription in each tissue (RPKM values).

Chromosome	Coordinate	Length	Family	Hypo-Dermis	Intestine	Neuron	Muscle	Comments
I	3424298	251	13_43_26	0.2	**1.95**	1.01	**2.76**	
II	1970474	311	46_32_6	0.20	**1.94**	0.33	0.45	Part of complete Helitron1_CE
II	2593754	249	46_32_6	0.25	**2.40**	0.66	**0.70**	Part of complete Helitron1_CE
II	3290648	136	10_25_40	0.06	1.62	**5.97**	0.35	
II	14993697	380	4_35_115	0.34	0.18	**5.33**	0.26	Part of Helitron1_CE
III	8344363	281	5_40_94	**2.84**	1.54	**5.05**	**1.68**	Part of HelitronY4_CE
IV	12314464	574	193_96_1	**1.43**	0.94	2.48	0.21	Unique satellite *
V	14156207	91	3_15_136	0.96	0.34	**5.40**	0.66	Part of HelitronY1A_CE
X	3693989	102	27_21_10	**2.12**	0.26	0.32	0.46	
X	15811201	80	9_20_46	**4.33**	**3.47**	4.74	**0.67**	
X	17441768	6016	234_45_1	0.59	0.02	0.11	0.02	Long, unique satellite

The four satellites with the highest transcription in each tissue are shown in bold. Transcription of the added terminal bases contributes to the transcription in several cases, in particular for the shortest satellites in the table. * Near this satellite was found another irregular satellite with the same 96 nt repeat [13].

**Table 2 ijms-24-02970-t002:** Main features of satellites in Fam_46_32_6 and the associated Helitron regions.

Orientation	15 nt Satellite		32 nt Satellite (Fam_4_35_115)		Conserved Region (nt) 5′UTR	Transposon Gene		31 nt Satellite (Fam_46_32_6)		Conserved Region(nt) 3′UTR
	Length (nt)	Coordinate	Length (nt)	Coordinate		Length (nt)	Name	Length (nt)	Coordinate	
-	76	857683	836	856800	590	4555	Y46B2A.2	361	851256	674
-	66	1969653	762	1968841	905	4874	ZK250.9	962	1962052	683
-	66	1976948	286	1976612	1894	3885	ZK250.10	311	1970474	640
-	81	1985898	826	1985022	904	4874	Y16E11A.2	1427	1977769	640
+	76	2002012	351	2002110	917	4874	F59H6.5	1179	2008283	679
-	94	2600074	224	2599830	927	4852	F33H12.6	249	2593754	683

## Data Availability

The sequence of all tandem repeats and a list of all tandem repeat families and their members are available in the Appendix A.

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
