# Peer review of "Unique Features of Satellite DNA Transcription in Different Tissues of *Caenorhabditis elegans"

_ijms, 2023, doi:10.3390/ijms24032970_

Round 1
Reviewer 1 Report
The paper by Subirana and Messeguer presents the results of RNA-seq analysis in different tissues of C. elegans in order to evaluate satellite DNA expression. They have found several general features that are exhibited and offer insight into transcription of the genome part that is for long period time considered to be silent, but nowadays more and more evidence is found for their functional role. Presented analysis are technically sound, the paper is well structured and the main point is understandable. However, there are still some open questions and shortcomings in the way of presenting results that need to be addressed.
Major concerns
1. Introduction is missing additional information about current state in the field of satellite transcription. Line 30-31 states “the transcription of tandem repeats or satellites has not been studied in detail”. However, there are several recent studies showing satellite expression in different organisms and it would be nice if the authors include some of the major discoveries.
2. I would kindly ask authors to provide details on how were the RNA-seq libraries they used prepared. Are they representing total RNA or are they enriched in mRNA? As this is an important fact for the study of satellite transcription it would be nice if the authors add somewhere comment on this topic.
3. Supplementary Data S1 contains several tables in different tabs with almost no description. I somewhat understand what the authors wanted to show but these tables should be numbered and have titles so they can be appropriately referred to within text.
4. It would be much easier to follow the results if the authors gave satellites names with numbers, (1-534) as currently they are mentioned as X: 17441768 for example (line 61).
5. Authors did not show or comment the genome abundance of investigated satellites. It would be nice to see if their expression correlates with satellite amount or not in order to rule out passive expression due to gene proximity.
6. Figure 1 needs improvement. Panel B is shown before panel A, image quality should be increased, please enlarge numbers and text on images so that they are visible or remove some if they are not necessary for understanding (this also applies for Figure 2 and Figure 4).
7. Supplementary Data S2 contains 533 image files (shouldn’t it be 534 as that is the total number of investigated satellites?) which is hard to track unless we want to look at specific satellite. That would be much easier if the authors renamed files so it contains satellite name, repeat unit length, and maybe average expression level. This would be one of the possible solutions, even though this amount of files does not seem to be completely appropriate way of presenting transcription results.
8. Supplementary tables S1, S2, and S3 are just briefly mentioned in Materials and Methods section (line 275). They could be also mentioned in the text of the results within described content they are referring to, as some parts lack the presentation of the specified results.
9. Figure 4. should be positioned below the text that is referring to it. Authors could add panel A and B mark to be more consisted with the other figure. Additionally, add somewhere in the figure or bellow, name of the family that these satellites belong to.
10. It would benefit any reader if the authors would add in Materials and Methods section at least one sentence summarizing satellite determination procedure (e.g. program used, main parameters…) after referring to previously described procedure (line 273-274).
11. The authors stated that they have used 27 SRR RNA-seq files from four tissues (line 284-285). Are those replicates of different stages? If yes, please comment, especially how you used them and was there consistency between them.
12. Regarding the normalization and sorting RNA-seq hits I have a question what are the lateral 250 nt added to each side of the satellite and why are they used? Are they neighboring regions found in the genome or some random sequences?
Minor comments
Line 79. Sentence “This is somewhat surprising, since the repeats in each satellite have a similar sequence.” is a bit confusing. What is meant by “repeats in each satellite”? Maybe different repeat units from the same satellite distributed in the genome, or?
Line 86. Referral to satellite families Fam_84_79_3 and Fam_4_35_115 is explained in the text but currently not visible in Figure 1B. Please include in figure which presented satellites belong to that family.
Line 133. What is, or how was “a normal average transcription” calculated?
Line 188-190. Please reformulate the end of this sentence as it is confusing.
Line 203. Text is in smaller font in this part.
Line 215. Missing punctuation at the end of a sentence.
Line 285. C. elegans should be in italic.
Author Response
Answer to Reviewer 1. In bold
The paper by Subirana and Messeguer presents the results of RNA-seq analysis in different tissues of C. elegans in order to evaluate satellite DNA expression. They have found several general features that are exhibited and offer insight into transcription of the genome part that is for long period time considered to be silent, but nowadays more and more evidence is found for their functional role. Presented analysis are technically sound, the paper is well structured and the main point is understandable. However, there are still some open questions and shortcomings in the way of presenting results that need to be addressed.
The suggestions of the reviewers have been very helpful to improve the presentation of our work. Following their suggestions we have added several paragraphs to clarify our methods:
* A more detailed introduction.
* Satellites and their families. A new section 2.1 in Results and an extended Section 4.1 in Methods.
* About the RNA-seq data. In Sections 2.2 and 4.2 we present a description of our Supplementary Data S1, which now includes Tables S4-S8.
* A description of how figures were prepared, at the end of Section 4.2. Note that the reproduction of the figures in the docx document is not perfect. The figures provided separately have a much better quality.
Major concerns
- Introduction is missing additional information about current state in the field of satellite transcription. Line 30-31 states “the transcription of tandem repeats or satellites has not been studied in detail”. However, there are several recent studies showing satellite expression in different organisms and it would be nice if the authors include some of the major discoveries. Corrected.
- I would kindly ask authors to provide details on how were the RNA-seq libraries they used prepared. Are they representing total RNA or are they enriched in mRNA? As this is an important fact for the study of satellite transcription it would be nice if the authors add somewhere comment on this topic.Corrected.
- Supplementary Data S1 contains several tables in different tabs with almost no description. I somewhat understand what the authors wanted to show but these tables should be numbered and have titles so they can be appropriately referred to within text. Corrected.
- It would be much easier to follow the results if the authors gave satellites names with numbers, (1-534) as currently they are mentioned as X: 17441768 for example (line 61). This would be confusing.
- Authors did not show or comment the genome abundance of investigated satellites. It would be nice to see if their expression correlates with satellite amount or not in order to rule out passive expression due to gene proximity. The number of satellites measures their genome abundance. We have only studied intergenic satellites, in general they are not in contact with genes.
- Figure 1 needs improvement. Panel B is shown before panel A, image quality should be increased, please enlarge numbers and text on images so that they are visible or remove some if they are not necessary for understanding (this also applies for Figure 2 and Figure 4). Figure 1 is improved. Enlargement will make comparisons very difficult.
- Supplementary Data S2 contains 533 image files (shouldn’t it be 534 as that is the total number of investigated satellites?) which is hard to track unless we want to look at specific satellite. That would be much easier if the authors renamed files so it contains satellite name, repeat unit length, and maybe average expression level. This would be one of the possible solutions, even though this amount of files does not seem to be completely appropriate way of presenting transcription results. Missing image has been added. Nomenclature of figures corrected.
- Supplementary tables S1, S2, and S3 are just briefly mentioned in Materials and Methods section (line 275). They could be also mentioned in the text of the results within described content they are referring to, as some parts lack the presentation of the specified results. Corrected.
- Figure 4. should be positioned below the text that is referring to it. Authors could add panel A and B mark to be more consisted with the other figure. Additionally, add somewhere in the figure or bellow, name of the family that these satellites belong to. Position of figure has been changed. Family names are given in the caption.
- It would benefit any reader if the authors would add in Materials and Methods section at least one sentence summarizing satellite determination procedure (e.g. program used, main parameters…) after referring to previously described procedure (line 273-274).Corrected.
- The authors stated that they have used 27 SRR RNA-seq files from four tissues (line 284-285). Are those replicates of different stages? If yes, please comment, especially how you used them and was there consistency between them. These questions are answered in Reference 11.
- Regarding the normalization and sorting RNA-seq hits I have a question what are the lateral 250 nt added to each side of the satellite and why are they used? Are they neighboring regions found in the genome or some random sequences?. An explanation is given.
Minor comments. All of them have been corrected or clarified.
Line 79. Sentence “This is somewhat surprising, since the repeats in each satellite have a similar sequence.” is a bit confusing. What is meant by “repeats in each satellite”? Maybe different repeat units from the same satellite distributed in the genome, or?
Line 86. Referral to satellite families Fam_84_79_3 and Fam_4_35_115 is explained in the text but currently not visible in Figure 1B. Please include in figure which presented satellites belong to that family.
Line 133. What is, or how was “a normal average transcription” calculated?
Line 188-190. Please reformulate the end of this sentence as it is confusing.
Line 203. Text is in smaller font in this part.
Line 215. Missing punctuation at the end of a sentence.
Line 285. C. elegans should be in italic.

Reviewer 2 Report
This paper covers levels of DNA transcription in different tissues in C. elegans. It would be an interesting topic to discuss but the following points need to be addressed to improve it.
I missed a more detailed introduction. The introduction is only based on 5 lines. It could be improved by talking about the definitions and organization of the genome and repetitive DNA sequences, where DNA satellite is placed, the roles and influences of repetitive DNA in the genome, insights in other organisms and further focusing on C. elegans, transposons and Helitron transposons, …
But my main concern is about procedures. How many replicates (biological and technical) in the RNA-seq analysis has been performed? I would like to see graphs that show averages.
Also, I think it is important to stablish a threshold for the signal. Moreover including positive and negative regions for transcription. It would give an idea the level of transcription in these regions. In this sense, it would show if these peaks of signal in these regions is just background or method noise. Please, explain this point. In addition, double checking and high reproducibility using quantitative PCR of some areas tested, including known external controls, is strongly recommended.
Other points:
Lines 43-50. Please, explain more in detail the analysis of transcription, for example thresholds applied for “some transcription” or high transcription. Use positive (exons) and negative controls of transcription regions. The concept Helitron appears without previous explanation (it is explained shortly at line 162). Extend the explanation of different observations.
The quality of Figures 1, 2 and 4 is extremely low. I cannot see or distinguish anything at all. Text says that “high resolution figures are given in Supplementary Data S2” but I don’t see the point of having an additional high-resolution figure for a non-visible main figure. Numbers and text must be higher.
Please check the text. The writing of text is not precise. Some examples are C. elegans (with space in between) (Lines 34, 163, 309) or Line 115, …, and this could also be translated into figures: panel B located upper than A.
Line 72. Satellites of the same family may be found both in introns and in intergenic regions. Define “may be”. Statistics.
Line 74. Transcription is usually higher in neurons. Define “usually”. Statistics.
Line 78. Would it be possible to add a line below the transcription profile to define where the satellite (or areas…) begins and ends? The spikes cannot be related to anything.
Line 82 and 83. Impossible to distinguish this patter in Figure 1 or 2. Perhaps using arrows?
Similar reasons in line 85.
Lines 123 to 125. Members? Short repeats? Please describe “members” or “short repeats” in detailed here or in the introduction to make easier the reading.
Please do a reorganization of the document and the results.
Round 2
Reviewer 1 Report
The authors have improved the manuscript by including some of the suggestions from the review but there are still some issues that should be resolved.
1. There are certain advances in the introduction but there should be more background about information that is mentioned. For example, main focus of the paper are the satellites of C. elegans and there should be some details about them. Are they already described somewhere, what is precisely known from other studies, is the genome rich with them (e.g. Line 42 „...we decided to study the abundant satellites...“ and line 48 „We had previously studied in great detail the satellites in C.elegans“).
2. There is no much improvement in figure quality within main text compared to first submission. In Supplementary Data S2 they are present in high quality unlike the ones included in manuscript. I suggest using one of lossless compression, such as LZW and increasing the font (that would solve „...resolution is diminished.“ in line 123). In current state it is inappropriate to call Figure 1B high resolution image as no text or numbers are visible at all.
3. Use of replicates seem a major concern. As there are 27 datasets corresponding to 4 tissues I currently understand that the authors added up the values for each tissue. This is actually opposite of the standard method for RNAseq analysis where the replicates are needed to interpret the results appropriately. Is there a specific reason for using this approach?
4. The authors should present satellite genome abundance somewhere, especially for ones with the highest transcription (Table 1). Their listed copy number within Table S1 is not suitable for viewing as there is 229 pages in the document.
5. Tables in Supplementary Data S1 should have titles with sufficient description of data.
6. Sentence from the abstract „Our demonstration that some satellite RNAs are transcribed adds a new family of non-coding RNAs, a new element in the world of RNA interference, with new paths for the control of mRNA translation.“ seems a bit exaggerated. Especially the part about adding new family of non-coding RNAs as transcription is satDNA has been shown before. Same is true for the almost the identical sentence from the discussion (Line 359-360).
Minor comments
Line 47. Referring to replicates as 27 experiments sounds weird
Line 57. „...but they are not perfect.“ This can be omitted
Figure 1. Position of the letter A should be above the first panel
Line 76. Missing punctuation
Line 93. Add space between sentences
Line 178. Two punctuations
Line 181. „Family“ should not be in capitalized letter
Line 215. should be ...near THE non-coding RNAs...
Line 220. Which figure is referred to
Table 2. There is a large space between words in third column
Line 305. Add space in „The 32nt satellite...“
Line 309. „...our informatics tools...“ what tools are being referred to here
Line 314. Should it be „Also the average repeat sequence thus DOES not determine the rate...“
Author Response
Reviewer 1. Answers in bold
The authors have improved the manuscript by including some of the suggestions from the review but there are still some issues that should be resolved.
We are very thankful to the reviewer for valuable suggestions which certainly improve the paper.
- There are certain advances in the introduction but there should be more background about information that is mentioned. For example, main focus of the paper are the satellites of C. elegans and there should be some details about them. Are they already described somewhere, what is precisely known from other studies, is the genome rich with them (e.g. Line 42 „...we decided to study the abundant satellites...“ and line 48 „We had previously studied in great detail the satellites in C.elegans“). A new paragraph has been added in the introduction.
- There is no much improvement in figure quality within main text compared to first submission. In Supplementary Data S2 they are present in high quality unlike the ones included in manuscript. I suggest using one of lossless compression, such as LZW and increasing the font (that would solve „...resolution is diminished.“ in line 123). In current state it is inappropriate to call Figure 1B high resolution image as no text or numbers are visible at all. New improved figures 1 and 2 have been included in the manuscript. The reviewer should also look at the figures which are submitted separately, as pdf or png. Including them into the main text as .docx diminishes their quality.
- Use of replicates seem a major concern. As there are 27 datasets corresponding to 4 tissues I currently understand that the authors added up the values for each tissue. This is actually opposite of the standard method for RNAseq analysis where the replicates are needed to interpret the results appropriately. Is there a specific reason for using this approach?. Each dataset corresponds to a different sample of adult tissue. The datasets for each tissue gave consistent results, as shown in Fig. 1B in reference 10. Therefore we added up the values for each tissue as reported in Table S5. Further comments have been added in Section 4.2
- The authors should present satellite genome abundance somewhere, especially for ones with the highest transcription (Table 1). Their listed copy number within Table S1 is not suitable for viewing as there is 229 pages in the document. We do not quite understand this question. Table S1 gives the properties of individual satellites, its content has been now simplified. Each satellite is unique, it occurs only once in the genome. The frequency of related satellites is described by their families. The overall frequency of satellites is now discussed in the new paragraph in the introduction.
- Tables in Supplementary Data S1 should have titles with sufficient description of data. Corrected
- Sentence from the abstract „Our demonstration that some satellite RNAs are transcribed adds a new family of non-coding RNAs, a new element in the world of RNA interference, with new paths for the control of mRNA translation.“ seems a bit exaggerated. Especially the part about adding new family of non-coding RNAs as transcription is satDNA has been shown before. Same is true for the almost the identical sentence from the discussion (Line 359-360). These sentences have been eliminated
Minor comments
Line 47. Referring to replicates as 27 experiments sounds weird. Not changed; we think it is Ok as it is.
Line 57. „...but they are not perfect.“ This can be omitted. Not changed; we think it is Ok as it is.
Figure 1. Position of the letter A should be above the first panel. Figure has been modified.
Line 76. Missing punctuation. Corrected.
Line 93. Add space between sentences. Corrected.
Line 178. Two punctuations. Corrected.
Line 181. „Family“ should not be in capitalized letter Corrected.
Line 215. should be ...near THE non-coding RNAs... Corrected.
Line 220. Which figure is referred to. Corrected.
Table 2. There is a large space between words in third column. Corrected.
Line 305. Add space in „The 32nt satellite...“ Corrected.
Line 309. „...our informatics tools...“ what tools are being referred to here. Corrected.
Line 314. Should it be „Also the average repeat sequence thus DOES not determine the rate...“ Corrected
Reviewer 2 Report
My doubts have been resolved, thank you.
But the authors indicate that "we do not think that it is necessary to present very large figures, although the compression diminishes their quality"
I leave this to a publisher's decision. I think they need to increase the size of the names and axis numbers in the figures as I can't make out anything at all in the pdf provided. But maybe it is due to compression as authors say. The editor will have access to the final quality version.
Overall, I consider publication.
Author Response
Answer in bold
My doubts have been resolved, thank you.
But the authors indicate that "we do not think that it is necessary to present very large figures, although the compression diminishes their quality"
I leave this to a publisher's decision. I think they need to increase the size of the names and axis numbers in the figures as I can't make out anything at all in the pdf provided. But maybe it is due to compression as authors say. The editor will have access to the final quality version.
Overall, I consider publication.
The reviewer considers publication; he is only worried by the quality of the figures. We have improved them.
Round 3
Reviewer 1 Report
I thank the authors for implementing the reviewers' suggestions. The introduction now includes sufficient background of the current field, and the methods and results are presented much clearly.
Overall, the manuscript is suitable for acceptance in my opinion.